# A Terpyridine-Fe^2+^-Based Coordination Polymer Film for On-Chip Micro-Supercapacitor with AC Line-Filtering Performance

**DOI:** 10.3390/polym13071002

**Published:** 2021-03-24

**Authors:** Hongxing Wang, Feng Qiu, Chenbao Lu, Jinhui Zhu, Changchun Ke, Sheng Han, Xiaodong Zhuang

**Affiliations:** 1School of Chemical and Environmental Engineering, Shanghai Institute of Technology, Haiquan Road 100, Shanghai 201418, China; wang_hx1910@163.com; 2Frontiers Science Center for Transformative Molecules, The Meso-Entropy Matter Lab, The State Key Laboratory of Metal Matrix Composites, Shanghai Key Laboratory of Electrical Insulation and Thermal Ageing, School of Chemistry and Chemical Engineering, Shanghai Jiao Tong University, Shanghai 200240, China; castle@sjtu.edu.cn (C.L.); zhujinhui1109@sjtu.edu.cn (J.Z.); 3School of Mechanical Engineering, Shanghai Jiao Tong University, Shanghai 200240, China; kechangchun@sjtu.edu.cn

**Keywords:** bis(terpyridine)-Fe complex, coordination polymer film, interfacial polymerization, redox activity, micro-supercapacitor

## Abstract

The preparation of redox-active, ultrathin polymer films as the electrode materials represents a major challenge for miniaturized flexible electronics. Herein, we demonstrated a liquid–liquid interfacial polymerization approach to a coordination polymer films with ultrathin thickness from tri(terpyridine)-based building block and iron atoms. The as-synthesized polymer films exhibit flexible properties, good redox-active and narrow bandgap. After directly transferred to silicon wafers, the on-chip micro-supercapacitors of TpPB-Fe-MSC achieved the high specific capacitances of 1.25 mF cm^−2^ at 50 mV s^−1^ and volumetric energy density of 5.8 mWh cm^−3^, which are superior to most of semiconductive polymer-based micro-supercapacitor (MSC) devices. In addition, as-fabricated on-chip MSCs exhibit typical alternating current (AC) line-filtering performance (−71.3° at 120 Hz) and a short resistance–capacitance (RC) time (0.06 ms) with the electrolytes of PVA/LiCl. This study provides a simple interfacial approach to redox-active polymer films for microsized energy storage devices.

## 1. Introduction

In the past decade, electrochemical energy storage devices have drawn much attention under the fast development of nanotechnology and the demand for clean and renewable energy [1]. The miniaturization of electrochemical electronics, such as batteries and supercapacitors, would become an important part of prospective electronic devices with mobile, wearable or implantable properties [2,3]. Among them, on-chip micro-supercapacitors (MSCs) have not only fast charge–discharge rate, high power density and cycling stability but also exhibit alternating current (AC) line-filtering ability [4,5,6]. For the fabrication of on-chip MSCs, the free-standing material films that scale up to centimeters are highly desirable [7,8,9]. To meet this requirement, the liquid–air or liquid–liquid interfacial polymerization approach has been widely used to prepare 2D polymer films, including covalent organic frameworks (COFs) [10,11], metal-organic frameworks (MOFs) [12], conjugated polymers [13,14] and conductive polymers [8,15]. However, the synthesis of these materials often suffers from the time-consuming process and harsh reaction condition by using strong acidic or oxidizing agents, such as catalysts. Thus, development of new method toward polymer films is essential to enable their application in on-chip MSCs.

Coordination polymers usually are constructed by complexion reaction of ligand-containing building blocks with various metal atoms without catalysts [16]. Currently, various conjugated monomer and metal atoms have been incorporated into the skeleton of coordination polymers, showing a great deal of advantages, like controllable optical/electric properties, good processability and thermal stability [17,18]. For example, terpyridine-metal (Tpy-Mn^+^) complex shows the excellent redox properties; thus, these Tpy-Mn^+^-based coordination polymers could be applied in wide optoelectronic applications [19,20,21,22,23,24,25]. Higuchi et al. reported the preparation of asymmetric supercapacitors by spray-coating two coordination polymers onto an indium tin oxide glass substrate, which showed high areal capacitances of 1.5–2.0 mF cm^−2^ and volumetric energy density of 10−18 mWh cm^−3^ [26]. More importantly, making the coordination polymer films into a large size prepares them efficiently by using the interfacial polymerization method, rendering them desirable for potential applications in Li-ion batteries [27], photoelectric anodes [28], electrochromism [29,30] and photofunctional sensors [31]. Compared with layer-by-layer growth of coordination polymer on the Au interdigital electrodes, these free-standing coordination polymer films would be transferred easily to variable substrates (e.g., Si wafers, glass) for the construction of on-chip MSCs. Fabrication of coordination polymer film-based on-chip MSCs remains a challenge.

Herein, we demonstrate a facile liquid–liquid interfacial polymerization of tri(terpyridine)-based building blocks and iron atoms to produce a coordination polymer film. The fabricated polymer film, with ultrathin thickness of ~300 nm, could be transferred directly to a glass for producing on-chip MSCs through the conventional approach. Due to their redox properties, the as-prepared on-chip MSCs exhibit a high specific areal capacitance (1.25 mF cm^−2^) and volumetric energy density (5.8 mWh cm^−3^) at 50 mV s^−1^. In addition, this MSC also shows a favorable AC line-filtering performance (−71.3° at 120 Hz) and a short relaxation time of 0.06 ms. This kind of coordination polymer film offers a new prospect toward fabrication of miniaturized electrochemical energy storage devices.

## 2. Materials and Methods

### 2.1. Materials

4-Formylphenylboronic acid, 2-acetylpyridine, potassium hydroxide, ammonium hydroxide, tetrakis(triphenylphosphine)palladium [Pd(PPh_3_)_4_], tribromobenzene, Iron(II) tetrafluoroborate hexahydrate, silica gel (300–400 mesh) and all of the organic reagents (ethanol, chloroform, tetrahydrofuran, dichloromethane, acetonitrile) were purchased from Titan (Shanghai, China). Commercial reagents and dry solvents were used without further purification.

### 2.2. Instruments

^1^H and ^13^C nuclear magnetic resonance (NMR) spectroscopy was measured on a Bruker 500 (500 MHz for proton, 125 MHz for carbon) spectrometer (Bruker, Karlsruhe, Germany) with tetramethylsilane as the internal reference, using CDCl_3_ and CD_3_OD as solvents; Matrix-Assisted Laser Desorption/Ionization Time of Flight mass pectrometry was performed on autoflex speed TOF/TOF (Bruker, Karlsruhe, Germany); FT–IR Spectrometer (FTIR) was performed on a Spectrum 100 instrument (Perkin Elmer, Boston, MA, USA); X-ray photoemission spectroscopy (XPS) was performed on an AXIS UltraDLD instrument (Shimadzu, Kyoto, Japan) using Al Kα radiation as the X-ray source with sweeps of 2, 8, 5, 10, 14 for survey spectrum, N 1s, F 1s, B 1s and Fe 2p, respectively. The deconvolution of XPS spectra was done as following: 1) Shift C 1s of adventitious carbon to 284.7 eV; 2) Constrain peak area ratios; 3) Constrain full width at half maxima (FWHM) to be equal to each other for all deconvoluted peaks from photoemission spectra of the same element. Carbon (C) 1s peak (284.7 eV) was used as reference for calibration; Optical microscopy (OM) was acquired using Leica biological microscope DM400; scanning electron microscopy (SEM) was obtained on a Zeiss Ultra Plus (Jena, Germany) field Emission Scanning electron microscope at an electric voltage of 5 kV; atomic force microscopy (AFM) was performed on a Multimode Nanoscope IIIa atomic force microscope; ultraviolet-visible spectrophotometer (UV–Vis) was recorded on a Lamda 950 (PerkinElmer Co., Waltham, MA, USA); thermogravimetric analyses (TGA) was performed in nitrogen atmosphere from ambient temperature to 800 °C at the rate of 20 °C min^−1^ on a Discovery TGA550 instrument (TGA, TA, New Castle, DE, USA); The N_2_ adsorption/desorption sorption isotherm was measured on the Auto-sorb-iQA3200-4 sorption analyzer (Quantatech Co., Connor, NY, USA). Melt point (m.p.) was performed on a SGW X-4B melting point apparatus from ambient temperature to 300 °C at the rate of 10 °C min^−1^.

### 2.3. Synthesis Procedures

Synthesis of 4′-(2,2′:6′,2″-terpyridine)phenylboracic acid. NaOH powder (9.60 g, 240 mmol) was added to the ethanol (200 mL) solution of 4-formylphenylboric acid (6.0 g, 40 mmol) and 2-acetylpyridine (10.6 g, 88 mmol). After stirring at 25 °C for 10 h, NH_4_OH aqueous solution (150 mL, 28–30%) was added. After refluxed for 20 h, the solution was cooled to room temperature. After filtered and washed, the solid was purified by washing with CHCl_3_ as a white product in yield of 61% (8.7 g, 98.88% HPLC purity). m.p.: > 300 °C; ^1^H NMR (CD_3_OD, 500 MHz, ppm): *δ* 8.68–8.71 (*d*, 2H), 8.67 (*s*, 2H), 8.64–8.66 (*d*, 2H), 7.99–8.02 (*dt*, 2H), 7.72–7.78 (*d*, 4H), 7.47–7.49 (*dd*, 2H); ^13^C NMR (CD_3_OD, 125 MHz, ppm): *δ* 157.54, 156.91, 153.18, 150.01, 138.74, 135.47, 135.16, 125.98, 125.28, 123.00, 119.53; MALD–TOF MS (m/z): Calculated. for [C_21_H_16_BN_3_O_2_+H]^+^: 354.13. Found: 354.03.

Synthesis of 1,3,5-tri(4-(2,2′:6′,2″-terpyridine)phenyl)benzene. To a solution of tribromobenzene (189.6 mg, 602 μmol) and 4′-(2,2′:6′,2″-terpyridine)phenylboracic acid (960 mg, 2.71 mmol) in THF (60 mL), aqueous NaOH (30 mL, 1 M) was added. The solution was freeze-pump-thawed three times and backfilled with argon; then Pd(PPh_3_)_4_ (60 mg) was added. The mixture was refluxed for 2 days under argon. After cooling to 25 °C, the reaction solution was then filtered through a Brinell funnel, washed with THF, water and Et_2_O, respectively, to give a white powder (TpPB) with yield of 70% (420 mg, 99.99% HPLC purity). m.p.: > 300 °C; ^1^H NMR (CDCl_3_, 500 MHz, ppm): *δ* 8.84 (*s*, 6H), 8.76 (*m*, 6H), 8.70 (*d*, 6H), 8.09 (*d*, 6H), 7.97 (*s*, 3H), 7.93 (*d*, 6H), 7.90 (*m*, 6H), 7.37 (*m*, 6H); MALDI–TOF MS (m/z): Calculated. for [C_69_H_45_N_9_+H]^+^: 1000.38. Found: 1000.99.

Synthesis of TpPB-Fe. Liquid–liquid interface polymerization approach was used to synthesis of Tpy-Fe^2+^-based coordination polymer film. Strain 0.1 mM (20 mL) TpPB solution with CH_2_Cl_2_ solution before use. The TpPB solution was placed in a 40 mm diameter vial, then 10 mL of pure water was added to cover the TpPB solution in order to form the liquid–liquid interface. Fe(BF_4_)_2_•6H_2_O aqueous solution (50 mM, 10 mL) was then added slowly. After 24 h, TpPB-Fe was formed at the interface. The aqueous layer and organic layer were replaced with pure water and organic ethanol and CH_2_Cl_2_ to remove free monomers. Finally, TpPB-Fe film was transferred to silica wafer.

### 2.4. Fabrication of the MSC Device

For preparation of the MSC devices, fresh TpPB-Fe film was transferred onto a clean glass (glass//TpPB-Fe). Then, the Au layer was deposited on the surface of the film by using the Magnetron Sputtering System JCP350 (Beijing Techno Science Co., Ltd., Beijing, China) (glass//TpPB-Fe//Au). The interdigitated electrode was prepared through laser scribing (Laser Marking Machine, 50 W, Nanjing, China) (glass//TpPB-Fe//Au finger). Finally, the electrolyte was dropped on the surface of the electrode for overnight (glass//TpPB-Fe//Au finger@electrolyte). The electrolytes of PVA/LiCl, PVA/H_2_SO_4_ and 1-ethyl-3-methylimidazolium tetrafluoroborate ([EMIM][BF_4_]) were used according to the method described in our previous work [8].

### 2.5. Electrochemical Measurements

The electrochemical measurements were investigated using a CHI 660E electrochemical workstation. The electrochemical properties of TpPB-Fe were studied by two-electrode in-plane MSCs. The cyclic voltammogram (CV) was examined at the scan rate of 5–10,000 mV s^−1^. Electrochemical impedance spectroscopy (EIS) was employed in the frequency range from 0.01 Hz to 100 kHz at room temperature. The specific capacitance values of the TpPB-Fe-MSCs were calculated according to the following Equation (1):(1)$$C_{device}=\frac{1}{2\times v\times (V_{i}-V_{t})}\times \int_{V_{t}}^{V_{i}}I(V)dV$$
where C_device_ is denoted as the capacitance from TpPB-Fe film electrode; *ν* is the scan rate (V s^−1^); *V_i_* and *V_t_* are the integration potential window of CV curves and I(V) is the voltammetry discharge current. $\int_{V_{t}}^{V_{i}}I(V)dV$ is the integrated area from CV curves. The total surface area of the device (cm^2^) was 0.54 cm^2^. The configuration used contained active electrode and cross finger electrode gap in this work.

The areal specific capacitance (C_A_, mF cm^−2^) and volumetric specific capacitance (C_V_, F cm^−3^) were calculated from the cyclic voltammograms (CV) curves by Equations (2) and (3):(2)$$C_{A}=\frac{C_{device}}{A}$$
(3)$$C_{V}=\frac{C_{device}}{V}$$
where *A* and *V* are the total area and volume of the device, respectively. The electrochemical performance of the whole device shown in the Ragone plot was based on the volumetric stack capacitance from the galvanostatic charge/discharge (GCD) data. The specific volumetric energy densities (*E_V_*, Wh cm^−3^) and power densities (*P_V_*, W cm^−3^) were calculated from Equations of (4) and (5):(4)$$E_{V}=\frac{1}{2}\times C_{V}\times \frac{{\left(\Delta V\right)}^{2}}{3600}$$
(5)$$P_{V}=\frac{E_{V}}{\Delta t}\times 3600$$
where ∆*V* is the discharge voltage range and ∆t is discharge time (s). To investigate the AC line-filtering performance of the TpPB-Fe-MSCs on-chip micro-supercapacitors, the EIS measurement was performed. The specific capacitance of the micro-supercapacitors can be described by using *C*′(*f*) and *C*″(*f*) according to the Equation of (6) and (7):(6)$$C^{\prime }(f)=\frac{-Z^{\prime \prime }(f)}{2\prod fS{|Z(f)|}^{2}}$$
(7)$$C^{\prime \prime }(f)=\frac{-Z^{\prime }(f)}{2\prod fS{|Z(f)|}^{2}}$$

Cyclic voltammograms (CVs) were performed in a three-electrode cell at a scan rate of 50 mV s^−1^ in an anhydrous, nitrogen-saturated solution of 0.1 M tetrabutylammonium hexafluorophosphate (Bu_4_NPF_6_) acetonitrile solution, using platinum as work electrodes and Ag/Ag^+^ as a reference electrode. The onset oxidation potential (*E*_1/2_ ox) of ferrocene was −0.02 eV versus Ag/Ag^+^. The conduction band energy level was determined from the oxidation onset of CV data. The absolute energy level of redox potential of Fc/Fc^+^ against vacuum is −4.40 eV. The electrochemically determined bandgap (*E*_bg, CV_) is the difference between the starting potential of the copolymer during oxidation and reduction.

## 3. Results

### 3.1. Synthesis and Morphology

Starting from commercial 4-formylphenylboronic acid, the 4′-(2,2′:6′,2″-terpyridine) phenylboracic acid was prepared by condensation reaction with 2-acetylpyridine, according to previous work [32]. After the Suzuki reaction with 1,3,5-triborombenzene, the target monomer of 1,3,5-tri(4-(2,2′:6′,2″-terpyridine)phenyl) benzene (TpPB) was obtained successfully. The detailed information could be found in Appendix A. The fabrication of coordination polymer film (TpPB-Fe) with the topological structure (without counter anions) is shown in Figure 1. Owing to the high coordination terpyridine with transition-metal atoms, the liquid–liquid interfacial polymerization toward TpPB-Fe films in the circumstance of a dichloromethane (CH_2_Cl_2_) solution of TpPB as the lower layer and an Iron(II) tetrafluoroborate hexahydrate (Fe(BF_4_)_2_•6H_2_O) aqueous solution as the upper layer. A resultant purple, free-standing film was almost covering the whole liquid interface (Figure 2a), which could be transferred onto various substrates, like Si wafer. The topographic characteristics of the TpPB-Fe film were investigated by scanning electron microscopy (SEM), optical microscopy (OM) and atomic force microscopy (AFM). In Figure 2b, OM image shows the obtained coordination polymer film with a macroscopic lateral dimension (>1 cm^2^). The partial wrinkles and crack indicate the flexible nature of TpPB-Fe film, which were folded when it transferred onto the silica wafer. The SEM image demonstrates that this film has the uniform surface with the differential edged regions (Figure 2c). The energy-dispersive X-ray spectroscopy mapping (EDX-mapping) results confirm that Fe, N and C are homogenously distributed in TpPB-Fe (Figure 2d). AFM result reveals that the average thickness of TpPB-Fe film is ~300 nm (Appendix A). These results demonstrate that bottom-up coordination reaction of terpyridine groups with Fe(II) ions gives them the ultrathin and free-standing coordination polymer film.

### 3.2. Structural Characterization

The chemical structure of the TpPB-Fe film was evaluated by Fourier-transform infrared (FTIR) spectroscopy and X-ray photoelectron spectroscopy (XPS). In Figure 3a, the stretching vibration of pyridine skeleton in TpPB is located at 1586 cm^−1^, while an obvious shift of broad peak to higher energy is observed at 1601 cm^−1^ in TpPB-Fe, attributing to the formation of the low electronic deficient of C=N∙∙∙Fe structure [30]. In addition, the new peak at 1086 cm^−1^ is assigned to the stretching vibration of BF_4_^−^. The XPS survey spectrum of TpPB-Fe shows the existence of C, N, Fe, B and F elements (Appendix A). The Fe 2p core level spectrum for TpPB-Fe is deconvoluted into two peaks at 721.5 eV and 708.5 eV, assigning to the Fe 2p_1/2_ and 2p_3/2_ binding energies for Fe^2+^ (Figure 3b). However, the Fe^3+^ 2p signals at 724.0 eV and 710.5 eV were also found in TpPB-Fe, suggesting the partial Fe(III) ion from the oxidation of Fe(II) ion. The N 1s spectrum of TpPB-Fe could be assigned to N∙∙∙Fe (399.7 eV) and pyridine N (398.5 eV) (Figure 3c), confirming the successful the coordination reaction between terpyridines and Fe atoms. Furthermore, the B 1s and F 1s spectra of TpPB-Fe exhibited at 685.3 and 193.6 eV (Appendix A), which could be attributed to BF_4_^−^ counter anions. The atomic ratio of Fe:N:B:F is calculated to be 1:6.35:1.78:7.12, which is near to the theoretical values (Fe:N:B:F =1:6:2:8). The thermal stability of TpPB-Fe was studied by thermo gravimetric analysis (TGA) measurement. The decomposition temperature with 5% weight loss is over 200 °C, suggesting the excellent thermal stability of Tpy-Fe^2+^ complex and aromatic building block (Figure 3d). Moreover, the porous structure of TpPB-Fe films was investigated by nitrogen adsorption–desorption analysis. In Figure 3e, the isotherm of TpPB-Fe exhibits type IV nitrogen sorption with a hysteresis loop, demonstrating the mesoporous nature of this coordination polymer film (Figure 3f). The specific surface area from Brunauer–Emmett–Teller (BET) calculations is 25.3 m^2^ g^−1^, which is larger than reported coordination polymers film (9.94 m^2^ g^−1^) [27].

### 3.3. Electronic Properties

The electronic properties of TpPB-Fe were investigated by using photophysical processes. The UV–Vis spectrum of TpPB reveals that a strong absorption band appears at around 310 nm, which is assigned to the π−π* transitions of aromatic unit of TpPB [31]. After coordinating with Fe atoms, a slight redshift of π−π* transitions to 317 nm is found, attributing to π−π stacking of the building block. Furthermore, the TpPB-Fe also shows a new absorption peak at 581 nm in the visible region, which is the typical metal-to-ligand charge transfer of terpyridine−metal complex (Figure 4a) [26,30]. Generally, the value at the intersection of the two dashed red lines is the optical bandgap (*E*_bg, optical_) [32,33]. In the resulting *T*_auc_ plot of TpPB-Fe, *E*_bg, optical_ is 1.66 eV (Figure 4b). To investigate the electronic structures of the as-prepared polymer, the cyclic voltammetry (CV) in acetonitrile with the electrolyte of Bu_4_NPF_6_ was carried out. In Figure 4c, TpPB-Fe clearly shows the presence of excellent reversible oxidation redox behavior in the 0.8−1.2 V region, indicating the Fe^2+^-to -Fe^3+^ transition of coordination polymer [26], while the reduction process demonstrates the extended π-conjugated skeleton of the terpyridine-based building block for the delocalization of electron over the whole backbone [34]. Based on the onset value of the first oxidation potential, the conduction band (*E*_cb_) energy level of TpPB-Fe is derived to be −3.56 eV (Figure 4d). Accordingly, the valence band (*E*_vb_) energy level is calculated to be −5.22 eV from the equation *E*_bg, optical_ = *E*_cb_ − *E*_vb_. In addition, the *E*_bg, optical_ of TpPB-Fe also is similar to with that of *E*_bg, CV_ (1.72 eV), resulted from the onset value of reduction potential (Appendix A). These results demonstrate the semiconducting nature of TpPB-Fe [8].

### 3.4. Micro-Supercapacitor Applications

Given its rich porous structure, redox properties of Tpy-Fe^2+^ complex, semiconducting characteristic of polymer skeleton and free-standing film feature, TpPB-Fe would be a promising candidate for energy storage in on-chip MSCs. As shown in Figure 5a, the as-prepared TpPB-Fe film is transferred directly to a Si wafer, first. After deposition of Au, an electrode was prepared by laser etching, and interdigitated electrodes are drop-casted with various organic electrolytes to produce on-chip MSCs (TpPB-Fe-MSC). The information of electrode area calculation for TpPB-Fe-MSC is given in Figure 5b and Appendix A. The electrochemical performance of the TpPB-Fe-MSC was carried out by using cyclic voltammetry (CV) with scan rates ranging from 5 mV s^−1^ to 10 V s^−1^ (Figure 5c and Appendix A). At low scan rates (5 to 500 mV s^−1^), these CV curves show the pair of redox peaks, which are mainly derived from the redox properties of Tpy-Fe^2+^ complex. With the increasing scan rate, CV curves gradually become the electric double-layer capacitive (EDLC) behavior, indicating the ultrafast charge–discharge ability of the TpPB-Fe-MSC [35,36]. Calculated from the CV results, the areal capacitances (*C*_A_) of TpPB-Fe-MSC with electrolytes PVA/LiCl, [EMIM][BF_4_] and PVA/H_2_SO_4_, as the function of scan rate, are illustrated in Figure 5d. At the 5 mV s^−1^, the CA could reach 1.25, 0.29 and 0.55 mF cm^−2^, respectively, for electrolytes of PVA/LiCl, PVA/H_2_SO_4_ and [EMIM][BF_4_]. The highest capacitance performance of TpPB-Fe-MSC in PVA/LiCl is better than many reported MSCs, including grapheme-based MSCs (0.14 mF cm^−2^ at 5 mV s^−1^) [37], mPPy@GO MSCs (75.5 μF cm^−2^ at 10 mV s^−1^) [35], azulene-based coordination polymer MSCs (102 μF cm^−2^ at 50 mV s^−1^) [6], TTF-TCNQ/graphene MSCs (0.62 mF cm^−2^ at 10 mV s^−1^) [38], etc. Furthermore, the areal capacitance of TpPB-Fe-MSC with PVA/LiCl (0.82 mF cm^−2^ at 10 mV s^−1^) is about 2.5 and 5.5 times greater than that with [EMIM][BF_4_] (0.32 mF cm^−2^) and PVA/H_2_SO_4_ (0.15 mF cm^−2^), respectively, which might be the synergetic effect between material stability, electrolytic viscosity and ionic mobility. Furthermore, the volumetric capacitances (*C*_V_) for TpPB-Fe-MSC with PVA/LiCl, [EMIM][BF_4_] and PVA/H_2_SO_4_, at 5 mV s^−1^, are 41.7, 18.2 and 9.7 F cm^−3^, respectively. These performances are superior to most previously reported semiconductive, polymer-based MSCs (Appendix A).

Electrochemical impedance spectroscopy (EIS) was used for inquiring into the ion transportation ability and frequency response of TpPB-Fe-MSCs in various electrolytes [39]. In Figure 6a, the Nyquist plots of TpPB-Fe-MSC with PVA/LiCl and [EMIM][BF_4_] exhibit the almost straight line and the intersections at the Z’ axis. Such the pronounced capacitance behavior indicates the fast ion mobility in the TpPB-Fe-MSC [7,8]. For the electrolyte of PVA/H_2_SO_4_, the acidic electrolyte could have the reaction with the structure of coordination polymers, leading to its low ion mobility [40]. In the region of high frequency, the minimum equivalent series resistance (ESR) with PVA/LiCl, [EMIM][BF_4_] and PVA/H_2_SO_4_ is 13.8, 35.1 and 27.3 Ω, respectively, which results from different ionic radius and viscosity of electrolytes for the ion transportation ability between active materials and electrolytes. In Figure 6b, the phase angle of TpPB-Fe-MSC, based on PVA/LiCl, reaches −75° at frequencies of 500 Hz, illustrating that TpPB-Fe-MSC has 83% of the ideal capacitor [41]. Additionally, the characteristic frequencies (*f*_0_) at the phase angle of 45°, attributing to the equal of the resistive and capacitive impedance values [42], are 5007, 16,326 and 58,770 Hz in electrolytes of PVA/H_2_SO_4_, PVA/LiCl and [EMIM][BF_4_], respectively. Corresponding, the relaxation time constant *t*_0_ (*t*_0_ = 1/*f*_0_) of these devices is 0.2, 0.06 and 0.02 ms for PVA/H_2_SO_4_, PVA/LiCl and [EMIM][BF_4_], respectively, which are much lower than activated carbon-based MSCs (~200 ms) [43], graphene–CNT MSCs (~0.74 ms) [42], B/N-enriched semiconductive polymer MSCs (0.52 ms) [7], azulene-based coordination polymer MSCs (0.27 ms) [6], benzene-bridged polypyrrole MSCs (0.22 ms) [8], etc. In addition, the TpPB-Fe-MSC displays an impedance phase angle of −71.3° at a frequency of 120 Hz. These results demonstrate the high AC line-filtering performance of TpPB-Fe-MSC, which exhibits its potential for flexible electric devices.

Ragone plots calculated from the CV results indicate the overall electrochemical performance of TpPB-Fe-MSCs. As shown in Figure 6b, TpPB-Fe-MSC in PVA/LiCl exhibits the largest energy density of 5.8 mWh cm^−3^ at 0.1 W cm^−3^ and largest power density of 9.8 W cm^−3^ at 0.3 mWh cm^−3^. This outstanding performance of TpPB-Fe-MSC is also compared with those of the reported MSCs, including a commercial lithium battery [44], CNT–graphene carpets [45], carbon onion-like [44], PANI nanowires [46], *d*-Ti_3_C_2_T_x_ [4] and electrolytic capacitors [2]. Cycle stability is also crucial importance for on-chip MSCs. The long-term cycling test of TpPB-Fe-MSC, based on PVA/LiCl, was carried out for 2500 cycles at a current density of 500 mV s^−1^. After 2500 cycles, the shape of CV changes a little. The retention of capacitance is 92.2%, indicating the high stability performance of TpPB-Fe-MSC (Figure 6c). These results indicate that the as-prepared Tpy-Fe^2+^-based coordination polymer film as electrode material have potential for high-performance MSCs.

## 4. Conclusions

In summary, we developed a novel tri(terpyridine)-based coordination polymer film containing a large area size, an ultrathin thickness and a uniform surface by liquid–liquid interfacial polymerization approach with 1,3,5-tri(4-(2,2′:6′,2″-terpyridine)phenyl) benzene and Fe^2+^ ionic resource. Such a 2D coordination polymer film possessed many physical properties, containing flexible properties, good redox activity and narrow bandgap. After used for MSCs, the as-prepared TpPB-Fe film, based on the PVA/LiCl electrolytes, delivered an ultrahigh areal capacitance (1.25 mF cm^−2^), volumetric energy density (5.8 mWh cm^−3^) and promising AC line-filtering properties (−71.3° at 120 Hz) with short time constant (0.06 ms). This study provides a simple method for achieving a flexible polymer film with low bandgap and redox activity for fabrication of both energy storage and optoelectronic applications.

## Figures and Tables

**Figure 1 polymers-13-01002-f001:**
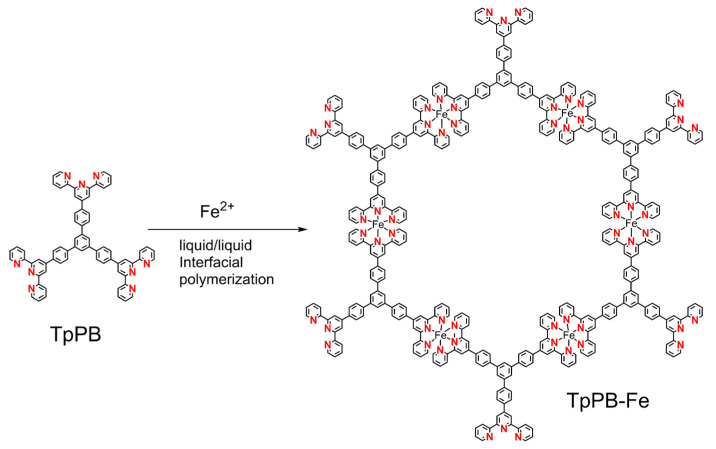
The structure of TpPB-Fe coordination polymer.

**Figure 2 polymers-13-01002-f002:**
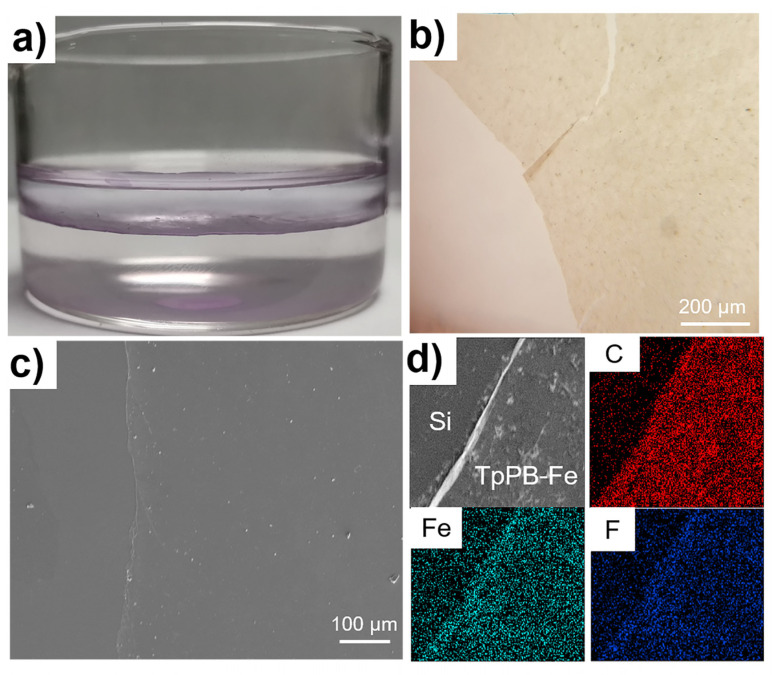
(**a**) The interfacial polymerization of TpPB-Fe coordination polymer between CH_2_Cl_2_/H_2_O; (**b**) Optical image, (**c**) SEM image, and (**d**) EDX-mapping of TpPB-Fe.

**Figure 3 polymers-13-01002-f003:**
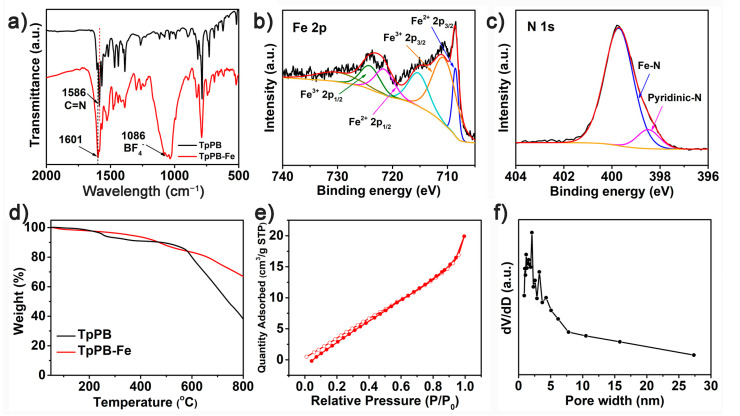
(**a**) The FTIR spectrum of TpPB-Fe coordination polymer between (**b**) Fe 2p and (**c**) N 1s XPS spectra of TpPB-Fe; (**d**) TGA curve of TpPB and TpPB-Fe; (**e**) Nitrogen adsorption/desorption isotherm of TpPB-Fe; (**f**) pore size distribution for TpPB-Fe.

**Figure 4 polymers-13-01002-f004:**
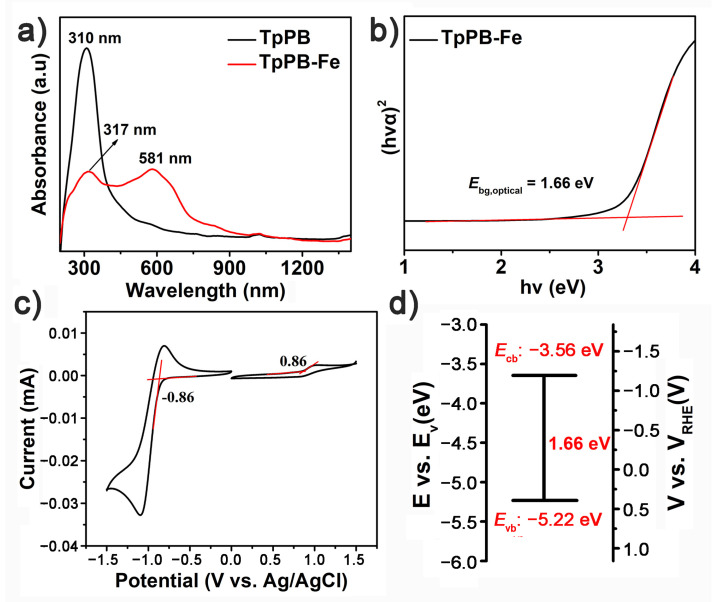
(**a**) The UV–Vis spectra of TpPB and TpPB-Fe; (**b**) bandgap (*E*_bg, optical_) of TpPB-Fe determined from the Kubelka–Munk-transformed reflectance spectra for TpPB-Fe; (**c**) cyclic voltammogram (CV) curve of TpPB-Fe; (**d**) band structure diagram for TpPB-Fe.

**Figure 5 polymers-13-01002-f005:**
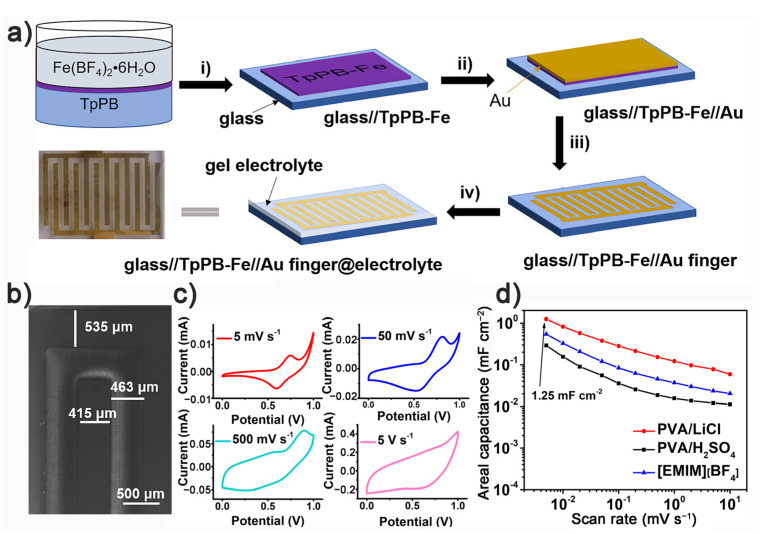
(**a**) Schematic illustration of TpPB-Fe films for on-chip MSCs on a silica substrate, (i) transferring TpPB-Fe films onto the glass substrate, (ii) sputtering an Au layer on TpPB-Fe film, (iii) laser scraping TpPB-Fe//Au heterolayer into interdigital pattern, iv) drop-casting of gel electrolyte on interdigitated electrode; (**b**) The size information of cross finger electrode; (**c**) CV curves of TpPB-Fe-MSC with PVA/LiCl gel electrolyte at scan rates of 5, 50, 500, 5000 mV s^−1^; (**d**) areal capacitance at scan rates from 5 to 10000 mV s^−1^ with the electrolytes of PVA/LiCl (red), [EMIM][BF_4_] electrolyte (blue) and PVA/H_2_SO_4_ (black), respectively.

**Figure 6 polymers-13-01002-f006:**
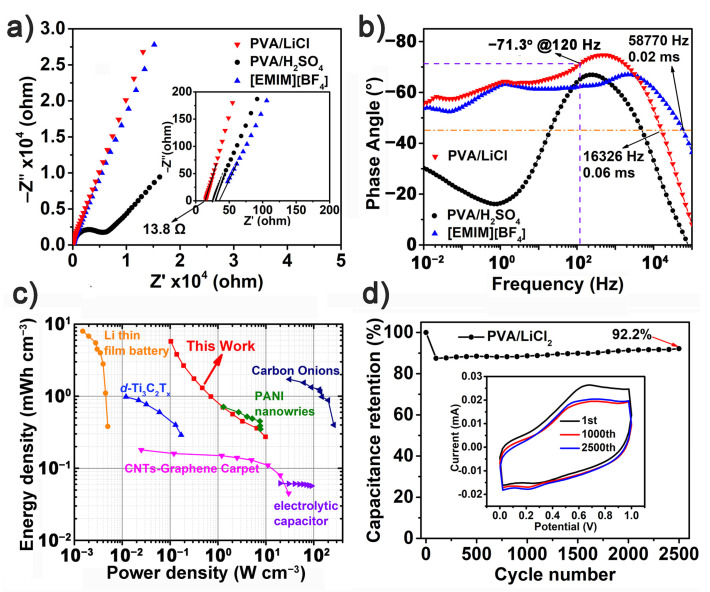
(**a**) Nyquist plots; (**b**) impedance phase angles on the frequency of TpPB-Fe-MSCs with PVA/LiCl (red), PVA/H_2_SO_4_ (black) and [EMIM][BF_4_] (blue); (**c**) Ragone plots for TpPB-Fe-MSCs with PVA/LiCl (red), PVA/H_2_SO_4_ (black) and [EMIM][BF_4_] (blue), comparing with other reported MSCs; (**d**) Cycling stability of TpPB-Fe-MSC with PVA/LiCl electrolyte (scan rate: 500 mV s^−1^), insert: the 1st, 1000th, 2500th CV curves of TpPB-Fe-MSC with PVA/LiCl electrolyte.

## Data Availability

The data presented in this study are available on request from the corresponding author.

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
