# Peer review of "A Terpyridine-Fe2+-Based Coordination Polymer Film for On-Chip Micro-Supercapacitor with AC Line-Filtering Performance"

_polymers, 2021, doi:10.3390/polym13071002_

Round 1

Reviewer 1 Report

This paper concerns the use of polymer coatings as ultrathin, flexible electrodes in flexible electronics. The work is interesting and the manuscript is well-structured. However, some critical methodological information is missing, as well as a few other minor additions/corrections.

EXPERIMENTAL:

  • Percentage purities of chemicals missing
  • NMR solvents and/or reference standards not stated.
  • Acquisition parameters, step sizes, etc., are missing for all instrumental methodologies. This needs to be added for all instruments. For example, XPS:
    • What was the pass energy?
    • What was the reference signal?
    • How many scans per reading?
    • How was the data analysed/data deconvolution done? Etc.
  • In the device fabrication section, even though the detailed procedure may have been given in previous work, the authors should at least offer an abbreviated summary of the procedure in this manuscript, for the benefit of the reader.

RESULTS:

  • BET: the authors refer to a reference value, but do not state it (page 6, line 229). That would be a helpful and valuable addition.
  • Figures: All the figures, while generally looking good are a little on the small side. It is hard to make out many of the details when everything is small and crammed together. Perhaps consider reorganising and expanding the images, at a higher resolution.

REFERENCES:

  • Inconsistencies in article title format. Some have capitalised words, others have a sentence-type format. Please pick a consistent style and use throughout.

Reviewer 2 Report

The manuscript “A Terpyridine-Fe2+-based Coordination Polymer Film for On-Chip Micro-Supercapacitor with AC Line-Filtering Performance” deals with the production of ultrathin polymer films, characterized by flexible properties, good redox-active and narrow bandgap. These films can be used as micro-sized energy storage devices. The work is interesting and well organized. Good results have been also obtained. Therefore, the publication is recommended; but after some revisions.

In particular:

- Please, define MSC the first time it appears in the Introduction.

- The state of the art in the Introduction can be enlarged adding other works related to alternative production techniques. For this purpose, see, for example, the work of Sarno et al., SC-CO2-assisted process for a high energy density aerogel supercapacitor: The effect of GO loading, Nanotechnology, 2017, 28, Article number 204001; etc..

- Please, add chemico-physical properties of chemicals.

- Please, remove bold from TpPB-Fe-MSC.

- Please, enlarge the graphs in Figure 6 since are difficult to be read.

- The morphology of the produced films can have an effect on the supercapacitor performance. Please, discuss this aspect.

Round 2

Reviewer 2 Report

The authors performed the modifications proposed by the Reviewer and improved the manuscript.